# Semantically-Guided Representation Learning for Self-Supervised Monocular Depth

**Vitor Guizilini**[1]  **Rui Hou**[1,2]  **Jie Li**[1]  **Rareş Ambruş**[1]  **Adrien Gaidon**[1]

[1]Toyota Research Institute (TRI)  [2]University of Michigan
{first.last}@tri.global  rayhou@umich.edu

## Abstract

Self-supervised learning is showing great promise for monocular depth estimation, using geometry as the only source of supervision. Depth networks are indeed capable of learning representations that relate visual appearance to 3D properties by implicitly leveraging category-level patterns. In this work we investigate how to leverage more directly this semantic structure to guide geometric representation learning, while remaining in the self-supervised regime. Instead of using semantic labels and proxy losses in a multi-task approach, we propose a new architecture leveraging fixed pretrained semantic segmentation networks to guide self-supervised representation learning via pixel-adaptive convolutions. Furthermore, we propose a two-stage training process to overcome a common semantic bias on dynamic objects via resampling. Our method improves upon the state of the art for self-supervised monocular depth prediction over all pixels, fine-grained details, and per semantic categories.[†]

## 1 Introduction

Accurate depth estimation is a key problem in computer vision and robotics, as it is instrumental for perception, navigation, and planning. Although perceiving depth typically requires dedicated sensors (e.g., stereo rigs, LiDAR), learning to predict depth from monocular imagery can provide useful cues for a wide array of tasks (Michels et al., 2005; Kendall et al., 2018; Manhardt et al., 2019; Lee et al., 2019). Going beyond supervised learning from direct measurements (Eigen et al., 2014), self-supervised methods exploit geometry as supervision (Guo et al., 2018; Pillai et al., 2019; Zou et al., 2018; Yang et al., 2017), therefore having the potential to leverage large scale datasets of raw videos to outperform supervised methods (Guizilini et al., 2019).

Although depth from a single image is an ill-posed inverse problem, monocular depth networks are able to make accurate predictions by learning representations connecting the appearance of scenes and objects with their geometry in Euclidean 3D space. Due to perspective, there is indeed an equivariance relationship between the visual appearance of an object in 2D and its depth, *when conditioned on the object's category*. For instance, a car 25 meters away appears smaller (on the image plane) than a car only 5 meters away but bigger than a truck 50 meters away. Current depth estimation methods either do not leverage this structure explicitly or rely on strong semantic supervision to jointly optimize geometric consistency and a semantic proxy task in a multi-task objective (Ochs et al., 2019; Chen et al., 2019), thus departing from the self-supervised paradigm.

In this paper, we explore how we can leverage semantic information to improve monocular depth prediction *in a self-supervised way*. Our **main contribution** is a novel architecture that uses a fixed pre-trained semantic segmentation network to guide geometric representation learning in a self-supervised monocular depth network. In contrast to standard convolutional layers, our architecture uses pixel-adaptive convolutions (Su et al., 2019) to learn semantic-dependent representations that can better capture the aforementioned equivariance property. Leveraging semantics may nonetheless introduce category-specific biases. Our **second contribution** is a two-stage training process where we automatically detect the presence of a common bias on dynamic objects (projections at infinity) and resample the training set to de-bias it. Our method improves upon the state of the art in self-supervised monocular depth estimation on the standard KITTI benchmark (Geiger et al., 2013), both on average over pixels, over classes, and for dynamic categories in particular.

---

[†]Source code and pretrained models are available on https://github.com/TRI-ML/packnet-sfm

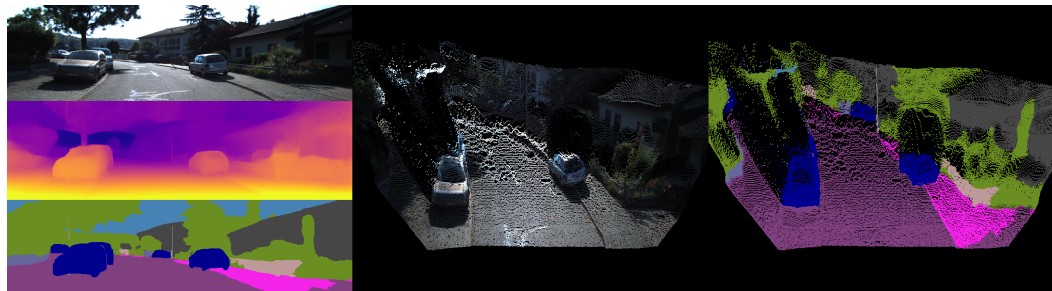

Figure 1: **Example of a pointcloud generated using our proposed** semantically-guided architecture, colored by RGB values from the input image and corresponding predicted semantic labels.

## 2 RELATED WORK

Since the seminal work of Eigen et al. (2014), substantial progress has been done to improve the accuracy of supervised depth estimation from monocular images, including the use of Conditional Random Fields (CRFs) (Li et al., 2015), joint optimization of surface normals (Qi et al., 2018), fusion of multiple depth maps (Lee et al., 2018), and ordinal classification (Fu et al., 2018). Consequently, as supervised techniques for depth estimation advanced rapidly, the availability of large-scale depth labels became a bottleneck, especially for outdoor applications. Garg et al. (2016) and Godard et al. (2017) provided an alternative self-supervised strategy involving stereo cameras, where Spatial Transformer Networks (Jaderberg et al., 2015) can be used to geometrically warp, in a differentiable way, the right image into a synthesized left image, using the predicted depth from the left image. The photometric consistency loss between the resulting synthesized and original left images can then be minimized in an end-to-end manner using a Structural Similarity term (Wang et al., 2004) and additional depth regularization terms. Following Godard et al. (2017) and Ummenhofer et al. (2017), Zhou et al. (2017) generalized this to the purely monocular setting, where a depth and a pose networks are simultaneously learned from unlabeled monocular videos. Rapid progress in terms of architectures and objective functions (Yin & Shi, 2018; Mahjourian et al., 2018; Casser et al., 2019; Zou et al., 2018; Klodt & Vedaldi, 2018; Wang et al., 2018; Yang et al., 2018) have since then turned monocular depth estimation into one of the most successful applications of self-supervised learning, even outperforming supervised methods (Guizilini et al., 2019).

The introduction of semantic information to improve depth estimates has been explored in prior works, and can be broadly divided into two categories. The first one uses semantic (or instance) information to mask out or properly model dynamic portions of the image, which are not accounted for in the photometric loss calculation. Güney & Geiger (2015) leveraged object knowledge in a Markov Random Field (MRF) to resolve stereo ambiguities, while Bai et al. (2016) used a conjunction of instance-level segmentation and epipolar constraints to reduce uncertainty in optical flow estimation. Casser et al. (2019) used instance-level masks to estimate motion models for different objects in the environment, and account for their external motion in the resulting warped image. The second category attempts to learn both tasks in a single framework, and uses consistency losses to ensure that both are optimized simultaneously and regularize each other, so the information contained in one task can be transferred to improve the other. For instance, Ochs et al. (2019) estimated depth with an ordinal classification loss similar to the standard semantic classification loss, and used empirical weighting to combine them into a single loss for optimization. Similarly, Chen et al. (2019) used a unified conditional decoder that can generate either semantic or depth estimates, and both outputs are used to generate a series of losses also combined using empirical weighting to generate the final loss to be optimized.

Our approach focuses instead on representation learning, exploiting semantic features into the self-supervised depth network by using a pretrained semantic segmentation network to guide the generation of depth features. This is done using pixel-adaptive convolutions, recently proposed in Su et al. (2019) and applied to tasks such as depth upsampling using RGB images for feature guidance. We show that different depth networks can be readily modified to leverage this semantic feature guidance, ranging from widely used *ResNets* (He et al., 2016) to the current state-of-the-art *PackNet* (Guizilini et al., 2019), with a consistent gain in performance across these architectures.

## 3   Self-Supervised Structure-From-Motion

Our semantically-guided architecture is developed within a self-supervised monocular depth estimation setting, commonly known as *structure-from-motion* (SfM). Learning in a self-supervised structure-from-motion setting requires two networks: a monocular depth model $f_D : I \rightarrow D$, that outputs a depth prediction $\hat{D} = f_D(I(p))$ for every pixel $p$ in the target image $I$; and a monocular ego-motion estimator $f_{\mathbf{x}} : (I_t, I_S) \rightarrow \mathbf{x}_{t \rightarrow S}$, that predicts the 6 DoF transformations for all $s \in S$ given by $\mathbf{x}_{t \rightarrow s} = \left( \begin{smallmatrix} \mathbf{R} & \mathbf{t} \\ \mathbf{0} & \mathbf{1} \end{smallmatrix} \right) \in$ SE(3) between the target image $I_t$ and a set of temporal context source images $I_s \in I_S$. In all reported experiments we use $I_{t-1}$ and $I_{t+1}$ as source images.

### 3.1   The Self-Supervised Objective Loss

We train the depth and pose networks simultaneously, using the same protocols and losses as described in Guizilini et al. (2019). Our self-supervised objective loss consists of an appearance matching term $\mathcal{L}_p$ that is imposed between the synthesized $\hat{I}_t$ and original $I_t$ target images, and a depth regularization term $\mathcal{L}_s$ that ensures edge-aware smoothing in the depth estimates $\hat{D}_t$. The final objective loss is averaged per pixel, pyramid scale and image batch, and is defined as:

$$\mathcal{L}(I_t, \hat{I}_t) = \mathcal{L}_p(I_t, \hat{I}_t) + \lambda_1 \, \mathcal{L}_s(\hat{D}_t) \tag{1}$$

where $\lambda_1$ is a weighting coefficient between the photometric $\mathcal{L}_p$ and depth smoothness $\mathcal{L}_s$ loss terms. Following Godard et al. (2017) and Zhou et al. (2017), the similarity between synthesized $\hat{I}_t$ and original $I_t$ target images is estimated using a Structural Similarity (SSIM) term (Wang et al., 2004) combined with an L1 loss term, inducing the following overall photometric loss:

$$\mathcal{L}_p(I_t, \hat{I}_t) = \alpha \, \frac{1 - \text{SSIM}(I_t, \hat{I}_t)}{2} + (1 - \alpha) \, \|I_t - \hat{I}_t\| \tag{2}$$

In order to regularize the depth in low gradient regions, we incorporate an edge-aware term similar to Godard et al. (2017). This loss is weighted for each of the pyramid levels, decaying by a factor of 2 on each downsampling, starting with a weight of 1 for the $0^{\text{th}}$ pyramid level.

$$\mathcal{L}_s(\hat{D}_t) = |\delta_x \hat{D}_t| e^{-|\delta_x I_t|} + |\delta_y \hat{D}_t| e^{-|\delta_y I_t|} \tag{3}$$

We also incorporate some of the insights introduced in Godard et al. (2018), namely auto-masking, minimum reprojection error, and inverse depth map upsampling to further improve depth estimation performance in our self-supervised monocular setting.

### 3.2   Depth and Pose Networks

Our baseline depth and pose networks are based on the *PackNet* architecture introduced by Guizilini et al. (2019), which proposes novel packing and unpacking blocks to respectively downsample and upsample feature maps during the encoding and decoding stages. This network was selected due to its state-of-the-art performance in the task of self-supervised monocular depth estimation, so we can analyze if our proposed architecture is capable of further improving the current state-of-the-art. However, there are no restrictions as to which models our proposed semantically-guided architecture can be applied to, and in Section 5.4 we study its application to different depth networks.

## 4   Semantically-Guided Geometric Representation Learning

In this section, we describe our method to inject semantic information into a self-supervised depth network via its augmentation with semantic-aware convolutions. Our proposed architecture is depicted in Figure 2 and is composed of two networks: a primary one, responsible for the generation of depth predictions $\hat{D} = f_D(I(p))$; and a secondary one, capable of producing semantic predictions. Only the first network is optimized during self-supervised learning; the semantic network is initialized from pretrained weights and is not further optimized. This is in contrast to the common practice of supervised (ImageNet) pretraining of depth encoders (Godard et al., 2018; Casser et al., 2019; Zou et al., 2018): here instead of fine-tuning from pre-trained weights, we preserve these secondary weights to guide the feature learning process of the primary depth network. Our approach also differs from learning without forgetting (Li & Hoiem, 2017) by leveraging fixed intermediate feature representations as a way to maintain consistent semantic guidance throughout training.

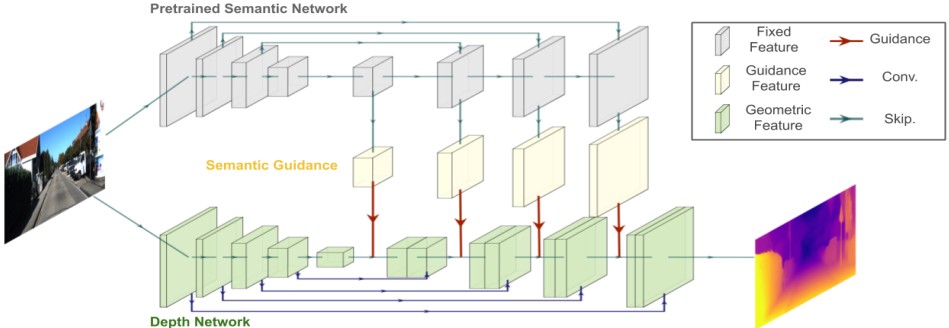

Figure 2: **Diagram of our proposed architecture** for self-supervised monocular depth estimation with semantically-guided feature learning. The semantic network is fixed and initialized from pretrained weights, while the depth network is trained end-to-end in a self-supervised way, including pixel-adaptive convolutions (Guidance) on its decoder to learn semantic-dependent geometric features.

## 4.1 SEMANTICALLY-GUIDED DEPTH FEATURES

We leverage the information from the pretrained semantic network in the depth network through the use of pixel-adaptive convolutions (Su et al., 2019). They were recently proposed to address some limitations inherent to the standard convolution operation, namely its translation invariance making it content-agnostic. While this significantly reduces the number of parameters of the resulting network, this might also lead to sub-optimal solutions under certain conditions important for geometric representation learning. For example, spatially-shared filters globally average the loss gradients over the entire image, forcing the network to learn weights that cannot leverage location-specific information beyond their limited receptive fields. Content-agnostic filters are unable to distinguish between different pixels that are visually similar (i.e. dark areas due to shadows or black objects) or generalize to similar objects that are visually different (i.e. cars with varying colors). In this work, we use pixel-adaptive convolutions to produce *semantic-aware depth features*, where the fixed information encoded in the semantic network is used to disambiguate geometric representations for the generation of multi-level depth features.

As shown in Figure 2, we extract multi-level feature maps from the semantic network. For each feature map, we apply a $3 \times 3$ and a $1 \times 1$ convolutional layer followed by Group Normalization (Wu & He, 2018) and ELU non-linearities (Clevert et al., 2016). These processed semantic feature maps are then used as guidance on their respective pixel-adaptive convolutional layers, following the formulation proposed in Su et al. (2019):

$$\mathbf{v}'_i = \sum_{j \in \Omega(i)} K(\mathbf{f}_i, \mathbf{f}_j) \mathbf{W}[\mathbf{p}_i - \mathbf{p}_j] \mathbf{v}_j + \mathbf{b} \tag{4}$$

In the above equation, $\mathbf{f} \in \mathcal{R}^D$ are processed features from the semantic network that will serve to guide the pixel-adaptive convolutions from the depth network, $\mathbf{p} = (x, y)^T$ are pixel coordinates, with $[\mathbf{p}_i - \mathbf{p}_j]$ denoting 2D spatial offsets between pixels, $\mathbf{W}_{k \times k}$ are convolutional weights with kernel size $k$, $\Omega_i$ defines a $k \times k$ convolutional window around $i$, $\mathbf{v}$ is the input signal to be convolved, and $\mathbf{b} \in \mathcal{R}^1$ is a bias term. $K$ is the kernel used to calculate the correlation between guiding features, here chosen to be the standard Gaussian kernel:

$$K(\mathbf{f}_i, \mathbf{f}_j) = \exp\left(-\frac{1}{2}(\mathbf{f}_i - \mathbf{f}_j)^T \Sigma_{ij}^{-1} (\mathbf{f}_i - \mathbf{f}_j)\right) \tag{5}$$

where $\Sigma_{ij}$ is the covariance matrix between features $\mathbf{f}_i$ and $\mathbf{f}_j$, here chosen to be a diagonal matrix $\sigma^2 \cdot I_D$, with $\sigma$ as an extra learnable parameter for each convolutional filter. These kernel evaluations can be seen as a secondary set of weights applied to the standard convolutional weights, changing their impact on the resulting depth features depending on the content stored in the guiding semantic features. For example, the information contained in depth features pertaining to the sky should not be used to generate depth features describing a pedestrian, and this behavior is now captured as a larger distance between their corresponding semantic features, which in turn produces smaller weights for that particular convolutional filter. Note that the standard convolution can be considered a special case of the pixel-adaptive convolution, where $\forall\, ij, K(\mathbf{f}_i, \mathbf{f}_j) = 1$.

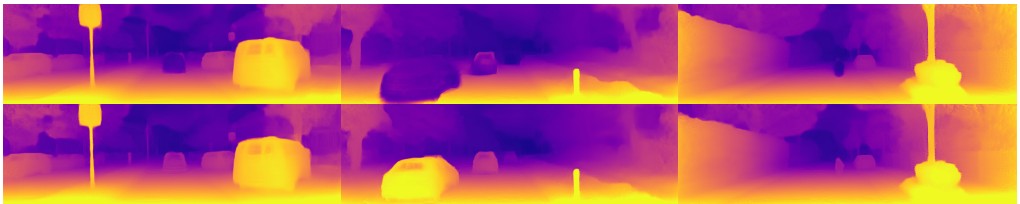

Figure 3: **Qualitative results of our proposed two-stage training** to address the infinite depth problem. Top images were obtained evaluating the first-stage depth network, and bottom images were obtained using the second-stage depth network, trained with a filtered dataset.

## 4.2 SEMANTIC GUIDANCE NETWORK

As the secondary network used to provide semantic guidance for the generation of depth features, we use a Feature Pyramid Network (FPN) with *ResNet* backbone (Lin et al., 2017). This architecture has been shown to be efficient for both semantic and instance-level predictions towards panoptic segmentation (Kirillov et al., 2019; Li et al., 2018; Xiong et al., 2019; Porzi et al., 2019). While our proposed semantically-guided architecture is not restricted to any particular network, we chose this particular implementation to facilitate the future exploration of different sources for guidance information. Architectural details follow the protocols described in Li et al. (2018), and unless mentioned otherwise the same pretrained model was used in all reported experiments. The semantic network is assumed fixed, pretrained on a held out dataset different than the raw data used for self-supervised learning, i.e. we do not require any semantic ground truth on the target dataset.

## 4.3 TWO-STAGE TRAINING

One well-known limitation of the self-supervised photometric loss is its inability to model dynamic objects, due to a static world assumption that only accounts for camera ego-motion (Godard et al., 2018; Casser et al., 2019). A resulting common failure mode is the *infinite depth* problem, which is caused by the presence of objects moving at the same speed as the camera. This typically causes distinct holes in the predicted depth maps, with arbitrarily large values where these objects should be. This severely hinders the applicability of such models in real-world applications, particularly for automated driving, where the ability to detect and properly model dynamic objects is crucial. Moreover, this limitation may be further accentuated in our proposed semantically-guided architecture, as the infinite depth problem occurs mostly on dynamic categories (i.e. cars and motorcycles) and the semantic-aware features may reinforce this bias.

We propose a simple and efficient two-stage training method to detect and remove this bias from the training set. In the first stage, we learn a standard depth network on all available training data. This network, exhibiting the infinite depth problem, is then used to resample the dataset by automatically filtering out sequences with infinite depth predictions that violate a basic geometric prior. We indeed find that depth predictions for pixels corresponding to the nearby ground plane are generally robust. This enables getting a coarse estimate of the ground plane using RANSAC and detecting the number of pixels whose predicted depth projects them significantly below the ground. If that number is above a threshold, then the corresponding image is subsequently ignored (we found a conservative threshold of 10 to work well in all our experiments, filtering out roughly 5% of the KITTI training dataset). During the second stage, we retrain the network on the subsampled dataset (from scratch to avoid the previous local optimum). As this subsampled dataset is de-biased, the network learns better depth estimates on dynamic objects. This process can be repeated, but we find that two stages are enough to remove any traces of infinite depth in our experiments, as shown in Figure 3.

## 5 EXPERIMENTAL RESULTS

## 5.1 DATASETS

We use the standard KITTI benchmark (Geiger et al., 2013) for self-supervised training and evaluation. More specifically, we adopt the training, validation and test splits used in Eigen et al. (2014) with the pre-processing from Zhou et al. (2017) to remove static frames, which is more suitable for

| Method | Superv. | Lower is Better | | | | Higher is Better | | |
|---|---|---|---|---|---|---|---|---|
| | | Abs Rel | Sq Rel | RMSE | RMSE$_{log}$ | $\delta < 1.25$ | $\delta < 1.25^2$ | $\delta < 1.25^3$ |
| Garg et al. (2016) | M | 0.152 | 1.226 | 5.849 | 0.246 | 0.784 | 0.921 | 0.967 |
| Zou et al. (2018) | M | 0.150 | 1.124 | 5.507 | 0.223 | 0.806 | 0.933 | 0.973 |
| Godard et al. (2017) | M | 0.141 | 1.186 | 5.677 | 0.238 | 0.809 | 0.928 | 0.969 |
| Zhan et al. (2018) | M | 0.135 | 1.132 | 5.585 | 0.229 | 0.820 | 0.933 | 0.971 |
| Godard et al. (2018) (R18) | M | 0.115 | 0.903 | 4.863 | 0.193 | 0.877 | 0.959 | 0.981 |
| Godard et al. (2018) (R50) | M | 0.112 | 0.851 | 4.754 | 0.190 | 0.881 | 0.960 | 0.981 |
| Guizilini et al. (2019) (MR) | M | 0.108 | 0.727 | 4.426 | 0.184 | 0.885 | 0.963 | 0.983 |
| Guizilini et al. (2019) (HR) | M | 0.104 | 0.758 | 4.386 | 0.182 | 0.895 | 0.964 | 0.982 |
| Casser et al. (2019) | S+Inst | 0.141 | 1.025 | 5.290 | 0.215 | 0.816 | 0.945 | 0.979 |
| Chen et al. (2019) | S+Sem | 0.118 | 0.905 | 5.096 | 0.211 | 0.839 | 0.945 | 0.977 |
| Ochs et al. (2019) | D+Sem | 0.116 | 0.945 | 4.916 | 0.208 | 0.861 | 0.952 | 0.968 |
| **Ours (MR)** | M+Sem | 0.102 | **0.698** | 4.381 | 0.178 | 0.896 | 0.964 | **0.984** |
| **Ours (HR)** | M+Sem | **0.100** | 0.761 | **4.270** | **0.175** | **0.902** | **0.965** | 0.982 |

Table 1: **Quantitative performance comparison of our proposed architecture** on KITTI for depths up to 80m. *M* refers to methods that train using monocular images, *S* refers to methods that train using stereo pairs, *D* refers to methods that use ground-truth depth supervision, *Sem* refers to methods that include semantic information, and *Inst* refers to methods that include semantic and instance information. *MR* indicates 640 x 192 input images, and *HR* indicates 1280 x 384 input images. Our proposed architecture is able to further improve the current state of the art in self-supervised monocular depth estimation, and outperforms other methods that exploit semantic information (including ground truth labels) by a substantial margin.

monocular self-supervised learning. This results in 39810 images for training, 4424 for validation, and 697 for evaluation. Following common practice, we pretrain our depth and pose networks on the CityScapes dataset (Cordts et al., 2016), consisting of 88250 unlabeled images. Unless noted otherwise, input images are downsampled to 640 x 192 resolution and output inverse depth maps are upsampled to full resolution using bilinear interpolation. Our fixed semantic segmentation network is pretrained on Cityscapes, achieving a mIoU of $75\%$ on the validation set.

## 5.2 IMPLEMENTATION DETAILS

We implement our models with PyTorch (Paszke et al., 2017) and follow the same training protocols of Guizilini et al. (2019) when optimizing our depth and pose networks. The initial training stage is conducted on the CityScapes dataset for 50 epochs, with a batch size of 4 per GPU and initial depth and pose learning rates of $2 \cdot 10^{-4}$ and $5 \cdot 10^{-4}$ respectively, that are halved every 20 epochs. Afterwards, the depth and pose networks are fine-tuned on KITTI for 30 epochs, with the same parameters and halving the learning rates after every 12 epochs. This fine-tuning stage includes the proposed architecture, where information from the fixed semantic network, pretrained separately, is used to directly guide the generation of depth features. There is no direct supervision at any stage during depth training, all semantic information is derived from the fixed secondary network.

When pretraining the semantic segmentation network, we use a *ResNet-50* backbone with Imagenet (Deng et al., 2009) pretrained weights and optimize the network for $48k$ iterations on the CityScapes dataset with a learning rate of $0.01$, momentum of $0.9$, weight decay of $10^{-4}$, and a batch size of 1 per GPU. Random scaling between $(0.7, 1.3)$, random horizontal flipping, and a crop size of $1000 \times 2000$ are used for data augmentation. We decay the learning rate by a factor of 10 at iterations $36k$ and $44k$. Once training is complete, the semantic segmentation network is fixed and becomes the only source of semantic information when fine-tuning the depth and pose networks on KITTI.

## 5.3 DEPTH ESTIMATION PERFORMANCE

Our depth estimation results are summarized in Table 1, where we compare our proposed architecture with other published works. From these results we can see that the introduction of semantically-guided geometric representation learning further improves upon the current state of the art in self-supervised monocular depth estimation from Guizilini et al. (2019), which served as our baseline. Our approach also outperforms other methods that leverage semantic information by a substantial margin, even those using ground-truth KITTI semantic segmentation and depth labels during train-

| Network | SEM | TST | Lower is Better | | | | Higher is Better | | | Class-Avg. |
|---|---|---|---|---|---|---|---|---|---|---|
| | | | Abs Rel | Sq Rel | RMSE | RMSE$_{log}$ | $\delta < 1.25$ | $\delta < 1.25^2$ | $\delta < 1.25^3$ | Abs Rel |
| ResNet-18 | | | 0.120 | 0.896 | 4.869 | 0.198 | 0.868 | 0.957 | 0.981 | 0.149 |
| | ✓ | | 0.117 | 0.854 | 4.714 | 0.191 | 0.873 | 0.963 | 0.981 | 0.139 |
| ResNet-50 | | | 0.117 | 0.900 | 4.826 | 0.196 | 0.873 | 0.967 | 0.980 | 0.144 |
| | ✓ | | 0.113 | 0.831 | 4.663 | 0.189 | 0.878 | 0.971 | 0.983 | 0.136 |
| PackNet | | | 0.108 | 0.727 | 4.426 | 0.184 | 0.885 | 0.963 | 0.983 | 0.132 |
| | ✓ | | 0.103 | 0.710 | 4.301 | 0.179 | 0.895 | 0.964 | 0.984 | 0.121 |
| | ✓ | ✓ | 0.102 | 0.698 | 4.381 | 0.178 | 0.896 | 0.963 | 0.984 | 0.117 |

Table 2: **Ablative analysis** of our semantic guidance (*SEM*) and two-stage-training (TST) contributions. The last column indicates class-average Abs. Rel. obtained by averaging all class-specific depth errors in Figure 4, while other columns indicate pixel-average metrics.

ing (Ochs et al., 2019). Furthermore, in Figure 5 we also present qualitative results showing the improvements in depth estimation generated by our proposed framework, compared to our baseline. Note how our semantically-guided architecture produces sharper boundaries and better object delineation, especially in structures further away or not clearly distinguishable in the input image.

## 5.4 ABLATIVE ANALYSIS

### 5.4.1 DIFFERENT DEPTH NETWORKS

To better evaluate our main contribution, we provide an ablative analysis showing how it generalizes to different depth networks. To this end, we consider two variations of the widely used *ResNet* architecture as the encoder for our depth network: *ResNet-18* and *ResNet-50* (the same pretrained semantic network was used in all experiments). Depth estimation results considering these variations are shown in Table 2, where we can see that our proposed semantically-guided architecture is able to consistently improve the performance of different depth networks, for all considered metrics.

### 5.4.2 CLASS-SPECIFIC DEPTH PERFORMANCE

To further showcase the benefits of our semantically-guided architecture, we also provide class-specific evaluation metrics, as shown in Figure 4. As we do not have ground-truth semantic segmentation for these images, we use the prediction of the semantic network to bin pixels per predicted category, and evaluate only on those pixels. From these results we can see that our proposed architecture consistently improves depth performance for pixels across all predicted classes, especially those containing fine-grained structures and sharp boundaries, e.g. poles and traffic signs.

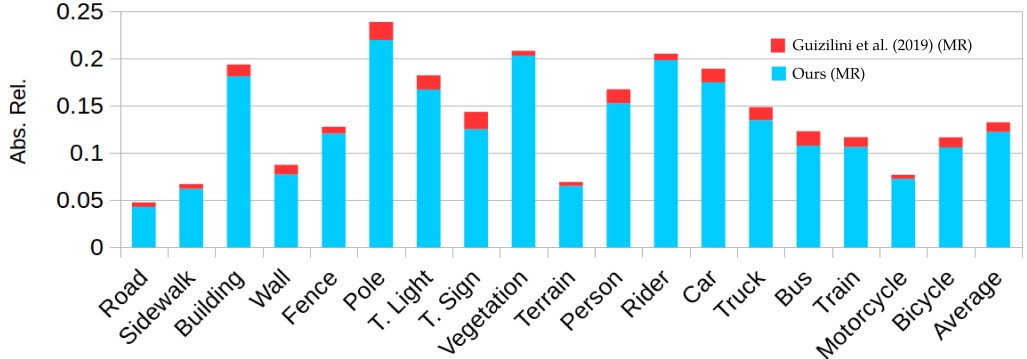

Figure 4: **Class-specific depth evaluation** for our proposed architecture (blue), relative to our baseline (red). The rightmost column indicates class-average depth metrics, obtained by averaging all individual classes. The introduction of semantically-guided features, in conjunction with our proposed two-stage training methodology to address the infinite depth problem, consistently improved depth results for all considered classes (lower is better).

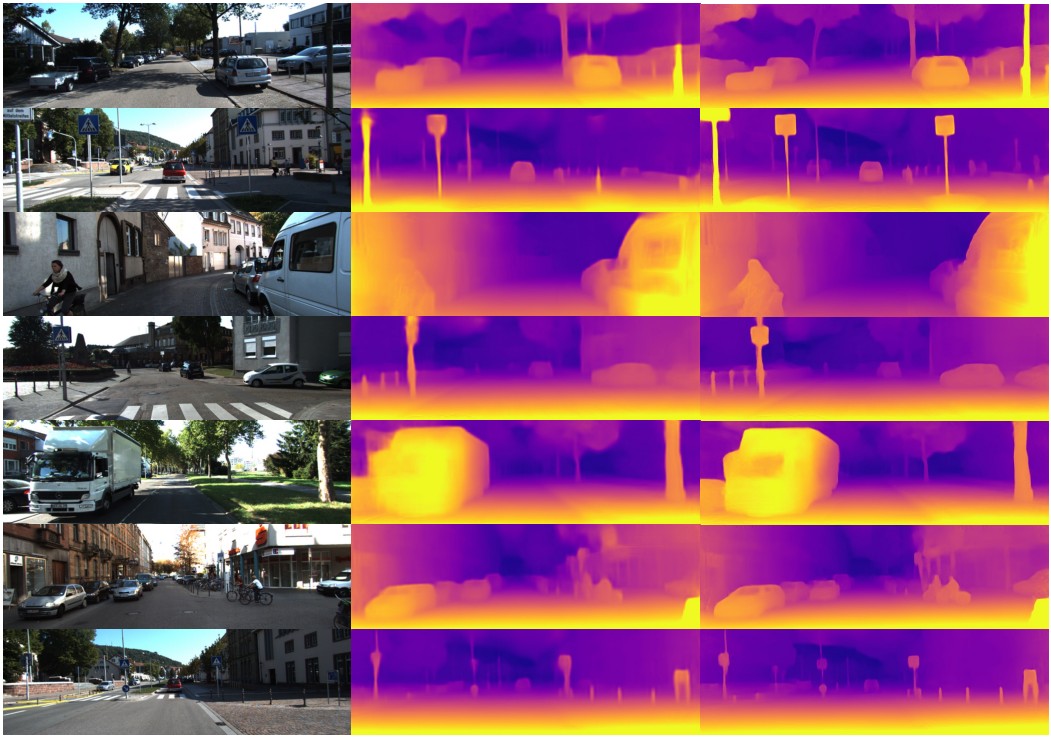

Figure 5: **Qualitative results of our proposed architecture.** The left, middle, and right columns show respectively input images, baseline predicted depth maps (Guizilini et al., 2019), and the depths maps obtained using our proposed architecture. Our semantic-aware depth network predicts sharper boundaries and fine-grained details on distant objects. The dotted lines indicate class-average errors, obtained by averaging all the class-specific depth errors.

We also measure the impact of our two-stage training process, which is expected to address the infinite depth problem in dynamic objects. Although we find the pixel-average difference in performance to not be significant (see Table 2), there is a significant improvement in class-average depth estimation, from 0.121 to 0.117 Abs-Rel. This is because the number of pixels affected by the infinite depth problem is vastly smaller than the total number of pixels. However, when considering class-average depth evaluation, the improvement over classes such as cars (0.200 to 0.177 Abs-Rel) and motorcycles (0.091 to 0.069) becomes statistically significant. This further exemplifies the importance of fine-grained metrics in depth evaluation, so these underlying behaviors can be properly observed and accounted for in the development of new techniques.

## 6    CONCLUSION

This paper introduces a novel architecture for self-supervised monocular depth estimation that leverages semantic information from a fixed pretrained network to guide the generation of multi-level depth features via pixel-adaptive convolutions. Our monodepth network learns semantic-aware geometric representations that can disambiguate photometric ambiguities in a self-supervised learning structure-from-motion context. Furthermore, we introduce a two-stage training process that resamples training data to overcome a common bias on dynamic objects resulting in predicting them at infinite depths. Our experiments on challenging real-world data shows that our proposed architecture consistently improves the performance of different monodepth architectures, thus establishing a new state of the art in self-supervised monocular depth estimation. Future directions of research include leveraging other sources of guidance (i.e. instance masks, optical flow, surface normals), as well as avenues for self-supervised fine-tuning of the semantic network.

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

## A  PRE-TRAINING THE SEMANTIC SEGMENTATION NETWORK

The introduction of a semantic segmentation network to the depth estimation task increases the depth estimation performance, however it also increases model complexity (e.g. number of trainable parameters). To investigate that the increased performance for the depth estimation task is indeed due to the semantic features encoded in the secondary network, we perform an in-depth analysis (summarized in Table 3) where we explore the impact of pre-training the semantic segmentation network before it is used to guide the generation of depth features. From these results we can see that the presence of semantic information encoded in the secondary network indeed leads to an increase in performance, and that fine-tuning this secondary network for the speficic task of depth estimation actually decreases performance.

In the first two rows an untrained semantic network is utilized, with only its encoder initialized from ImageNet (Deng et al., 2009) weights. Two different scenarios are explored: in the first one (D) only the depth network is fine-tuned in a self-supervised fashion, while in D+S both networks are fine-tuned together in the same way. As expected, using untrained features as guidance leads to significantly worse results, since there is no structure encoded in the secondary network and the primary network needs to learn to filter out all this spurious information. When both networks are fine-tuned simultaneously, results improve because now the added complexity from the secondary network can be leveraged for the task of depth estimation, however there is still no improvement over the baseline.

Next, the semantic network was pre-trained on only half of the CityScapes (Cordts et al., 2016) dataset (samples chosen randomly), leading to a worse semantic segmentation performance (validation mIoU of around 70% vs. 75% for the fully trained one). This partial pre-training stage was enough to enable the transfer of useful information between networks, leading to improvements over the baseline. Interestingly, fine-tuning both networks for the task of depth estimation actually hurt performance this time, which we attribute to forgetting the information contained in the secondary network, as both networks are optimized for the depth task. When the semantic network is pretrained with all of CityScapes (last two rows), these effects are magnified, with fine-tuning only the depth network leading to our best reported performance (Table 1) and fine-tuning both networks again leading to results similar to the baseline.

| Method | Pre-Train | Fine-Tune | Lower is Better | | | | Higher is Better | | |
|---|---|---|---|---|---|---|---|---|---|
| | | | Abs Rel | Sq Rel | RMSE | $RMSE_{log}$ | $\delta < 1.25$ | $\delta < 1.25^2$ | $\delta < 1.25^3$ |
| Baseline | — | D | 0.108 | 0.727 | 4.426 | 0.184 | 0.885 | 0.963 | 0.983 |
| Proposed | I | D+S | 0.116 | 0.847 | 4.751 | 0.192 | 0.879 | 0.960 | 0.981 |
| | I | D | 0.197 | 1.323 | 6.114 | 0.265 | 0.776 | 0.918 | 0.966 |
| | CS (1/2) | D+S | 0.109 | 0.737 | 4.389 | 0.185 | 0.884 | 0.962 | 0.982 |
| | CS (1/2) | D | 0.104 | 0.716 | 4.322 | 0.180 | 0.893 | 0.964 | 0.984 |
| | CS | D+S | 0.107 | 0.741 | 4.407 | 0.183 | 0.883 | 0.963 | 0.983 |
| | CS | D | **0.102** | **0.698** | **4.381** | **0.178** | **0.896** | **0.964** | **0.984** |

Table 3: **Analysis of the impact of pre-training** the semantic segmentation network. On the *Pre-Train* column, *I* indicates ImageNet (Deng et al., 2009) pretraining and *CS* indicates CityScapes (Cordts et al., 2016) pretraining, with *1/2* indicating the use of only half the dataset (samples chosen randomly). In the *Fine-Tune* column, *D* indicates fine-tuning the depth network and *S* indicates fine-tuning the semantic network (note that this is a self-supervised fine-tuning for the depth task, using the objective described in Section 3.1).

## B  UNCERTAINTY AND GENERALIZATION TO DIFFERENT OBJECTS

In a self-supervised setting, increasing the number of unlabeled videos used for depth training is expected to lead to an increasing specialization away from the domain in which the semantic network was pre-trained. This might result in harmful guidance if our method is not robust to this gap. However, our approach does not use semantic predictions directly, but rather the decoded features of the semantic network themselves, which represent general appearance information that should be more robust to this domain gap. To validate our hypothesis, we further explore the impact

of erroneous semantic information in the performance of our proposed semantically-guided depth framework. In Figure 6 we present qualitative results highlighting situations in which our pretrained semantic network failed to generate correct semantic predictions for certain objects in the scene, and yet our proposed framework was still able to properly recover depth values for that portion of the environment. These exemplify possible scenarios for erroneous semantic prediction.

- *Imprecise boundaries:* in the first row, we can see that the semantic segmentation network does not correctly detect the traffic sign, yet the semantically-guided depth network predicts its shape and depth accurately.

- *Wrong classification:* in the second row, the truck was mistakenly classified as partially "road" and "building", however our semantically-guided depth network was still able to properly recover its overall shape with sharp delineation that was not available from its semantic contour. A similar scenario happens in the same image, with "fence" being partially labeled as "bicycle".

- *Missing ontology:* there is no "trash can" class on the CityScapes ontology, however in the third row our semantically-guided depth network was able to correctly reconstruct such object even though it was classified as "fence", similarly to its surroundings.

- *Object Hallucination:* in the fourth row, the contour of a "person" was erroneously introduced in the image and correctly removed by our semantically-guided framework.

These examples are evidence that our proposed framework is able to reason over the uncertainty inherent to semantic classification, leveraging this information when accurate to achieve the results reported in this paper, but also discarding it if necessary to generate a better reconstruction according to the self-supervised photometric loss.

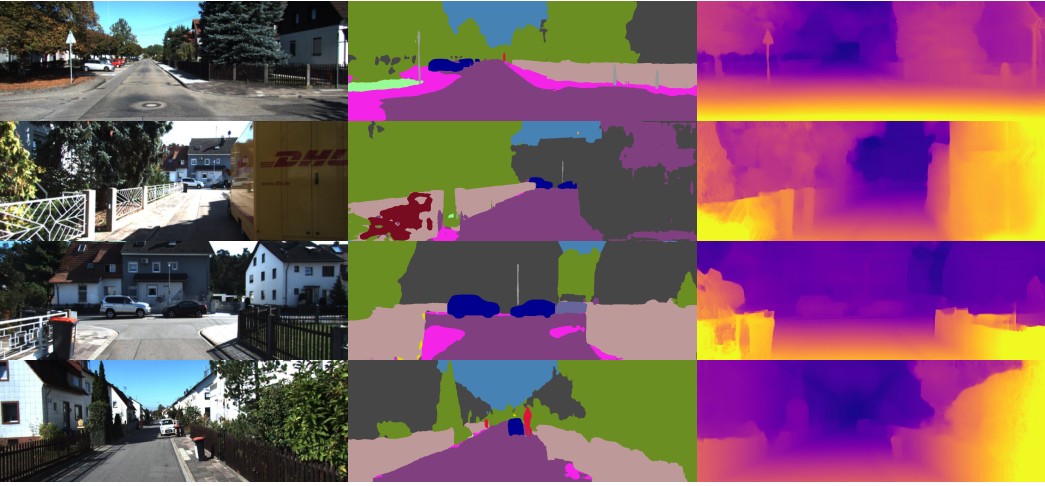

Figure 6: **Examples of erroneous semantic predictions** that still led to accurate depth predictions using our proposed semantically-guided depth framework.

## C  GENERALIZATION TO DIFFERENT DATASETS

In the previous sections, we show that our proposed framework is robust to a degraded semantic network, both by pretraining the semantic network with fewer annotated labels (Appendix A) and also by providing evidence that the depth network is able to reason over erroneous predictions to still generate accurate reconstructions (Appendix B). We now go one step further and analyze how our proposed semantically-guided framework generalizes to a dataset that was used neither during pre-training nor for fine-tuning. To this end, we evaluate our KITTI depth model on the recently released NuScenes dataset (Caesar et al., 2019). The official NuScenes validation split is used, containing 6019 images from the front camera with ground-truth depth maps generated by LiDAR reprojection. Results presented in Table 4 provide additional evidence that our method indeed results in generalization improvements, even on significantly different data from different platforms

and environments (Karlsruhe, Germany for KITTI vs Boston, USA and Singapore for NuScenes), outperforming the state of the art methods and our baseline (Guizilini et al., 2019).

| Method | Abs Rel | Sq Rel | RMSE | RMSE$_{log}$ | $\delta < 1.25$ | $\delta < 1.25^2$ | $\delta < 1.25^3$ |
|---|---|---|---|---|---|---|---|
| Godard et al. (2018) (R18) | 0.212 | 1.918 | 7.958 | 0.323 | 0.674 | 0.898 | 0.954 |
| Godard et al. (2018) (R50) | 0.210 | 2.017 | 8.111 | 0.328 | 0.697 | 0.903 | 0.960 |
| Guizilini et al. (2019) (MR) | 0.187 | 1.852 | 7.636 | 0.289 | 0.742 | 0.917 | 0.961 |
| **Ours (MR)** | **0.181** | **1.505** | **7.237** | **0.271** | **0.765** | **0.931** | **0.969** |

Table 4: **Generalization capability of different networks**, trained on both KITTI and CityScapes datasets and evaluated on the NuScenes (Caesar et al., 2019) dataset. Our proposed semantically-guided architecture is able to further improve upon the baseline from Guizilini et al. (2019), which only used unlabeled image sequences for self-supervised depth training.

