# OpenReview forum: "Semantically-Guided Representation Learning for Self-Supervised Monocular Depth"
_ICLR.cc/2020/Conference — Accept (Poster)_

### Official Review · AnonReviewer3 · 2019-10-20
**Official Blind Review #3**

**Rating:** 6

**Review:**

This work proposes to leverage a pre-trained semantic segmentation network to learn semantically adaptive filters for self-supervised monocular depth estimation. Additionally, a simple two-stage training heuristic is proposed to improve depth estimation performance for dynamic objects that move in a way that induces small apparent motion and thus are projected to infinite depth values when used in an SfM-based supervision framework. Experimental results are shown on the KITTI benchmark, where the approach improves upon the state-of-the-art.

Overview:

+ Good results
+ Doesn't require semantic segmentation ground truth in the monodepth training set

- Not clear if semantic segmentation is needed
- Specific to street scenes
- Experiments only on KITTI

The qualitative results look great and the experiments show that semantic guidance improves quantitative performance by a non-trivial factor. The qualitative results suggest that the results produced with semantic guidance are sharper and more detailed. However, it is not clear that using features from a pre-trained semantic segmentation network is necessary. The proposed technical approach is to use the pixel-adaptive convolutions by Su et. al. to learn content-adaptive filters that are conditioned on the features of the pre-trained semantic segmentation network. These filters could in principle be directly learned from the input images, without needing to first train a semantic segmentation network. The original work by Su et. al. achieved higher detail compared to their baseline by just training the guidance network jointly.  Alternatively, the guidance network could in principle be pre-trained for any other task. The main advantage of the proposed scheme is that the guidance path doesn't need to be trained together with the depth network. On the other hand, unless shown otherwise, we have to assume that the network needs to be pre-trained on some data that is sufficiently close to the indented application domain. This would limit the approach to situations where a reasonable pre-trained semantic segmentation network is available.

The proposed heuristic to filter some dynamic objects is very specific to street scenes and to some degree even to the KITTI dataset. It requires a dominant ground plane and is only able to detect a small subset of dynamic motion (e.g. apparent motion close to zero and object below the horizon). It is also not clear what the actual impact of this procedure is. Section 5.4.2 mentions that Abs. Rel decreases from 0.121 to 0.119, but it is not clear to what this needs to be compared to as there is no baseline in any of the other tables with an Abs. Rel of 0.121. Additionally, while the authors call this a minor decrease, the order of magnitude is comparable to the decrease in error that this method shows over the state-of-the-art (which the authors call statistically significant) and also over the baselines (c.f. Table 2). Can the authors clarify this?

Related to being specific to street scenes: The paper shows experiments only on the KITTI dataset. The apparent requirement to have a reasonable semantic segmentation model available, make it important to evaluate also in other settings (for example on an indoor dataset like NYU) to show that the approach works beyond street scenes (which is one of the in practice not so interesting settings for monocular depth estimation since it is rather easy to just equip cars with additional cameras to solve the depth estimation problem).

Need for a reasonable segmentation model: It is not clear in how far the quality of the segmentation network impacts the quality for the depth estimation task. What about the domain shift where the segmentation model doesn't do so well? Even if the segmentation result is not used directly, the features will still shift. How much would depth performance suffer?

Summary:
While the results look good on a single dataset, I have doubts both about the generality of the proposed approach as well as the need for the specific technical contribution.

=== Post rebuttal update ===
The authors have addressed many of my initial concerns and provided valuable additional experimental evaluations. While I'd like to upgrade my recommendation to weak accept, I strongly encourage the authors to provide additional experiments on different datasets (at least NYU).

**Experience Assessment:**

I have published one or two papers in this area.

**Review Assessment: Checking Correctness Of Derivations And Theory:**

N/A

**Review Assessment: Checking Correctness Of Experiments:**

I assessed the sensibility of the experiments.

**Review Assessment: Thoroughness In Paper Reading:**

I read the paper at least twice and used my best judgement in assessing the paper.

---

> ### Author Response · Authors · 2019-11-12
> **Response to Reviewer #3 (Part 3/3)**
>
> [Section 5.4.2 mentions that Abs. Rel decreases from 0.121 to 0.117, but it is not clear to what this needs to be compared to as there is no baseline in any of the other tables with an Abs. Rel of 0.121.]
>
> We apologize for the misunderstanding. What we are claiming and presenting evidence for is:
> Our semantically guided self-supervised depth estimation is our main contribution, significantly improving over the state of the art and unguided baseline on all metrics, esp. when averaging over semantic categories (0.121 vs 0.132 for the baseline);
> our two-stage training process fixes a rare but important catastrophic failure in practice (infinite depth), which results in further improvements on top of our main contribution (0.117 vs 0.121 class-average without two-stage training), although seemingly not significant when averaging over pixels (0.102 vs 0.103).
> These results are now clarified in the updated Table 2 and Figure 4, as well as properly discussed in Section 5.4.2.
>
>
> [Show that the approach works beyond street scenes (which is one of the in practice not so interesting settings for monocular depth estimation since it is rather easy to just equip cars with additional cameras to solve the depth estimation problem).]
>
> First, we agree with the reviewer that estimating generalization performance on other very different datasets would strengthen our already thorough experimental evaluation. As adding an entirely new benchmark represents a non-trivial amount of effort (and time), we will add NYUv2 experiments in the final version of the paper. In the meantime, we conducted an additional generalization experiment in Appendix C, where we evaluate how our KITTI model transfers to the NuScenes [6] dataset without any fine-tuning. Our model outperforms both our baseline and the previous state of the art (Godard et al 2018, Guizilini 2019) on all metrics. This provides additional evidence that our method indeed results in generalization improvements, even on significantly different data from different platforms and environments (Karlsruhe, Germany for KITTI vs Boston, USA and Singapore for NuScenes).
> Second, we believe that street scenes are not a small niche, but a major challenge and focus of the research community (as attested by the numerous related works we are citing in Section 2) and industry (e.g, for autonomous driving, delivery robots, augmented reality in urban environments, and many other applications). In particular, recent works on pseudo-lidar point-cloud generation [4, 5] are attempting to overcome a big performance gap between active range sensors and ubiquitous monocular cameras. Multi-camera setups require precise cross-sensor calibration to avoid compounding errors, and are not as widely used as single cameras (which are standard on all production cars today and on most mobile phones).
>
>
> [1] Huan Fu, Mingming Gong, Chaohui Wang, Kayhan Batmanghelich and Dacheng Tao. Deep Ordinal Regression Network for Monocular Depth Estimation. In Proceedings of the IEEE Conference on Computer Vision and Pattern Recognition (CVPR), 2018.
> [2] Clément Godard, Oisin Mac Aodha, Michael Firman, and Gabriel J. Brostow. Digging into self-supervised monocular depth prediction. In Proceedings of the International Conference on Computer Vision (ICCV), 2019.
> [3] Sudeep Pillai, Rares Ambrus, and Adrien Gaidon. Superdepth: Self-supervised, super-resolved monocular depth estimation. In Proceedings of the IEEE International Conference on Robotics and Automation (ICRA), 2019.
> [4] Yan Wang, Wei-Lun Chao, Divyansh Garg, BharathHariharan, Mark Campbell and Kilian Weinberger. Pseudo-LiDAR from Visual Depth Estimation: Bridging the Gap in 3D Object Detection for Autonomous Driving. In Proceedings of the IEEE Conference on Computer Vision and Pattern Recognition (CVPR), 2019.
> [5] Yurong You, Yan Wang, Wei-Lun Chao, Divyansh Garg, Geoff Pleiss, Bharath Hariharan, Mark Campbell and Killian Weinberger.Pseudo-LiDAR++: Accurate Depth for 3D Object Detection in Autonomous Driving. In arXiv:1906.06310
> [6] Holger Caesar, Varun Bankiti, Alex H. Lang, Sourabh Vora, Venice Erin Liong, Qiang Xu, Anush Krishnan, Yu Pan, Giancarlo Baldan, and Oscar Beijbom. NuScenes: A Multimodal dataset for autonomous driving. CoRR, 2019.

---

> ### Author Response · Authors · 2019-11-12
> **Response to Reviewer #3 (Part 2/3)**
>
> [The proposed heuristic to filter some dynamic objects is very specific to street scenes and to some degree even to the KITTI dataset.]
>
> The problem of "infinite depth" happens in scenes where dynamic agents move like the camera, which is frequent when navigating in structured environments. In particular, this is not limited to KITTI but ubiquitous in self-driving contexts (e.g., vehicles on the road driving at the speed limit), which is a major robotic application of depth estimation. This often manifests as a dataset bias that creates a wrong but stable local optimum in the self-supervised photometric objective (as it is coherent with its underlying projective geometry).
>
> We believe our proposed two-stage-training solution is original and interesting from a representation learning perspective, as it is neither an architectural, data augmentation, nor a loss improvement, but consists instead of automatically de-biasing the dataset thanks to a general structural inductive bias: objects do not go below the ground. The only assumption we are leveraging is that we can get a coarse estimate of the ground plane, which does not need to be "dominant" for classical robust estimation methods like RANSAC to get a good estimate.
>
> The impact of our two stage training on pixel-level metrics is minor, as our method corrects predictions for objects covering only a small part of the image. Nonetheless, it fixes a major "bug" of self-supervised monocular depth networks for downstream safety-critical applications like collision avoidance or object detection, as illustrated in Figure 3 and quantified in the per-class metrics (cf. the updated Table 2, class-average improvement from 0.121 to 0.117 Abs.Rel).
>
> Nonetheless, we agree that it is not the main contribution from our submission, although we believe it adds non-negligible value in an important use case. As far as we know, this is the first technique able to eliminate the infinite depth problem in an entirely self-supervised setting without the introduction of extra information. We also would like to note that the proposed two-stage-training does not require semantic information, and therefore can be equally applied to any self-supervised depth estimation framework.

---

> ### Author Response · Authors · 2019-11-12
> **Response to Reviewer #3 (Part 1/3)**
>
> [It is not clear that using features from a pre-trained semantic segmentation network is necessary. These filters could in principle be directly learned from the input images, without needing to first train a semantic segmentation network.]
>
> We agree with the reviewer that testing this hypothesis was missing from our ablative analysis. Therefore, we performed an additional experiment, added to Table 3 in Appendix A, in which we pretrain the semantic network only on ImageNet. This significantly degrades performance (0.197 vs 0.108 for the baseline vs 0.102 Abs.Rel. for our method). This can be partially mitigated when fine-tuning (self-supervised) the semantic and depth networks together, but it still degrades compared to the unguided baseline (0.116 vs 0.108). This confirms that, in practice, pretraining the semantic segmentation network is necessary, and our method can effectively transfer indirectly related semantic representations without overfitting.
>
>
> [Need for a reasonable segmentation model: It is not clear in how far the quality of the segmentation network impacts the quality for the depth estimation task.]
>
> We agree with the reviewer that, even though we are not directly using semantic predictions, a better semantic network should improve our model, and "unreasonably bad" semantic features (e.g., due to a large domain gap) will degrade our approach. Thus, it is important to estimate the relation between the quality (in the target domain) of the semantic network and our final self-supervised depth estimation performance. In particular, our guided feature learning approach needs to be robust to suboptimal guiding features to be widely applicable.
>
> Consequently, we added another experimental evaluation in Table 3 in Appendix A where we evaluate how depth estimation performance degrades with a worse semantic network. We trained our semantic network on only half of the CityScapes dataset (resulting in a significant 5% decrease in mIoU and qualitatively degraded predictions on KITTI), and use the same self-supervised learning protocol. We find only a minor performance degradation (0.104 vs 0.102 when using the fully trained semantic network). Furthermore, as reported above, even with no semantic information (i.e. just an ImageNet pretrained semantic network), our performance does not entirely collapse, although it goes below the baseline and strongly benefits from end-to-end finetuning.
>
> In addition, we provide new qualitative evidence in Appendix B Figure 6, showing that our depth network is robust to i) objects never seen by the semantic network (i.e. out of ontology), ii) pixels that have wrong semantic predictions, iii) objects hallucinated by the semantic network. Our previous results and these additional ones together confirm our guided feature learning approach is robust to a degradation in the pretrained semantic network, while effectively leveraging its additional information when useful, thus showing wide applicability (i.e. one can re-use a fixed pretrained semantic network across potentially dissimilar unlabeled target domains).

---

### Official Review · AnonReviewer1 · 2019-10-23
**Official Blind Review #1**

**Rating:** 6

**Review:**

The paper proposes a using pixel-adaptive convolutions to leverage semantic labels in self-supervised monocular depth estimation. The semantic features are predicted by a pretrained network rather than relying on a ground truth. Moreover, a two-stage training process in proposed in order to filter out images leading to erroneous SfM predictions. The method is evaluated with different networks on the KITTY dataset.

The paper is very well written and clear. The applications of per-pixel convolutions to this problem seems sound and the experimental validation seems satisfactory. I have however one main concern (1) and a few additional questions below:

1) While (Guizilini 2019) shows that using a larger set of unannotated videos and allows the self-supervised method to eventually outperform supervised methods, this study is not done here. This makes me question the applicability of the approach, as using large unlabelled videos would probably lead to noisy segmentations that could be unhelpful to the depth estimation. Showing an improvement over the supervised baseline would be a much stronger experimental validation, as for now it is difficult to know exactly why in which scenario this method should be used, rather than a supervised network or vanilla packnet.

2) I see that you obtain the same numbers in Table 2 / PackNet / row 1 as in (Guizilini 2019); I would like to confirm that you used exactly their self-objective loss, in all your experiments? I would suggest adding to section 3.1. the fact the fact that the loss is the same is in (Guizlini 2019), as a reader could assume that there is novelty in the loss formulation.

3) Have you tried fine-tuning the whole architecture including the semantic network end-to-end?


**Experience Assessment:**

I have published one or two papers in this area.

**Review Assessment: Checking Correctness Of Derivations And Theory:**

I carefully checked the derivations and theory.

**Review Assessment: Checking Correctness Of Experiments:**

I carefully checked the experiments.

**Review Assessment: Thoroughness In Paper Reading:**

I read the paper thoroughly.

---

> ### Author Response · Authors · 2019-11-12
> **Response to Reviewer #1**
>
> [It is difficult to know exactly why in which scenario this method should be used, rather than a supervised network or vanilla packnet.]
>
> We apologize for the lack of clarity in the text, which we updated in the new draft, as the scenario described by the reviewer is actually what we do. (Guizilini 2019) indeed showed that using unlabeled CityScapes videos improves the performance of self-supervised monocular depth estimation on KITTI to the point where it becomes competitive with fully supervised methods. We use exactly this model as our baseline, with the same training protocol as mentioned in Section 5.1. Our contribution lies in additionally leveraging the features of a fixed semantic network (pretrained on a separate dataset) to further improve the performance of the depth network, which is pre-trained in a self-supervised way on the same large unlabeled dataset as (Guizilini 2019). As the reviewer points out, because we do not use semantic labels during fine-tuning, this could lead to noisy segmentations on the unlabeled videos, which might hurt self-supervised learning at scale. However, as shown in the newly added Figure 6 in Appendix B, we find that our learned semantic guidance framework is able to leverage pretrained semantic features in a robust way, determining when their information should be used or discarded in order to further optimize the self-supervised photometric loss.
> Furthermore, in Table 1 we compare our proposed framework with the supervised method of Ochs et al. [1], which uses ground truth for both depth and semantic labels: we significantly outperform their results, even though our method is self-supervised.
> In conclusion, our proposed semantic guidance for self-supervised depth estimation improves on state-of-the-art self-supervised (including the baseline PackNet) and supervised methods (including with semantic and depth supervision). Furthermore, it is widely applicable, as it can leverage fixed semantic segmentation networks pretrained on different datasets and does not require semantic supervision in the target unlabeled dataset.
>
>
> [I would like to confirm that you used exactly their self-objective loss, in all your experiments.]
>
> We reproduced the state-of-the-art results of (Guizilini 2019) using the same network, loss, and settings, which allowed us to use their approach as our baseline. Any improvements generated by our proposed framework are due exclusively to our proposed contributions, i.e. our semantically-guided representation learning and two stage training process. We have added this information to Section 3.1.
>
>
> [Have you tried fine-tuning the whole architecture including the semantic network end-to-end?]
>
> We agree that exploring this further would yield interesting insights. To this end we have performed an additional analysis in Appendix A as suggested. Specifically, we study the effects of pre-training the semantic network on different amounts of data and the effects of fine-tuning both networks using only the self-supervised photometric loss. We observe that:
> When the semantic network is simply pre-trained on ImageNet and does not encode any dense semantic information, keeping it fixed significantly degrades depth estimation performance (0.197 vs 0.108 for the unguided baseline vs 0.102 for our method). Fine-tuning both networks in this case improves performance, but it still falls short of the unguided baseline performance (0.116 vs 0.108).
> When the semantic network is pre-trained on only half of the training set of CityScapes, it can still successfully guide the depth network, although the semantic predictions are degraded (-5% mIoU on Cityscapes). Final depth estimation performance improves a bit less, but the difference is relatively small (0.104 vs 0.102 when fully training the semantic network), confirming the robustness of our approach to a suboptimal semantic network.
> The previous observation holds as long as the semantic network is not fine-tuned. If both networks are fine-tuned, the overall performance indeed suffers and returns to values similar to the baseline (0.107 vs 0.108 for the unguided baseline vs 0.102 for our method), which we attribute to overfitting (the number of free parameters augmenting drastically) and eventual forgetting of the semantic information in the semantic network (as can be attested by the degraded semantic predictions we observed).
>
>
> [1] Matthias Ochs, Adrian Kretz, and Rudolf Mester. Sdnet: Semantically guided depth estimation network. In arXiv:1907.10659, 2019.

---

### Official Review · AnonReviewer2 · 2019-10-24
**Official Blind Review #2**

**Rating:** 3

**Review:**

The underlying problem considered in this manuscript is inferring depth from geometry of two dimensional images. The novelty here is to integrate a semantic classification model along with depth inference. It is a challenging problem and a neat idea to pursue. The paper is well-written and easy to follow.
However, the empirical work in the paper is not persuasive. Table 1 contains results whose significance is hard to judge. What does a RMSE difference of 2.3 mean in the context of depth estimation? Table 1 carries no uncertainty in results which is just not acceptable in a setting that has several sources of uncertainty. Similarly, Fig 4 shows there is advantage in the use of the pre-trained semantic network, it is not clear if this difference is significant. And I also think consideration should be given to the fact that in a deployment setting, new objects not previously seen in the semantic categories (UFO) may appear and one ought to understand if the semantic network might decrease performance (because of the unseen class). Hence I think which the idea advanced in the paper has merits, the manuscript is not really ready for publication.

**Experience Assessment:**

I have read many papers in this area.

**Review Assessment: Checking Correctness Of Derivations And Theory:**

I assessed the sensibility of the derivations and theory.

**Review Assessment: Checking Correctness Of Experiments:**

I assessed the sensibility of the experiments.

**Review Assessment: Thoroughness In Paper Reading:**

I read the paper at least twice and used my best judgement in assessing the paper.

---

> ### Author Response · Authors · 2019-11-12
> **Response to Reviewer #2 (Part 2/2)**
>
> [Fig 4 shows there is an advantage in the use of the pre-trained semantic network, it is not clear if this difference is significant.]
>
> Tables 1 and 2 present our depth results averaged over all pixels in the image, and we note an improvement from 0.108 to 0.102 Abs. Rel (i.e. a 6% relative improvement). As mentioned above, typical improvements over the state of the art, as reported by the community, are around 5%. To further clarify the impact of our contribution, we added the last column of Table 2: “Class-Average Abs Rel”, which represents the average per-class performance of our model compared to the baseline (rightmost bar in Figure 4). This evaluation is less biased (by dominant classes like road or vegetation), and it is in line with the typical evaluation protocol for semantic segmentation. We note a much more significant improvement when averaging class-based metrics, rather than pixel-based metrics: from 0.132 to 0.117 Abs. Rel.
> Using Figure 4 for further introspection, we note that our method improves depth performance on all semantic classes. In particular, these improvements are more pronounced for dynamic classes (i.e. cars and pedestrians) and thin structures (i.e. poles and traffic signs), which are particularly challenging to model in a self-supervised setting. Moving objects are not taken into consideration in the photometric loss, which relies on a static world assumption, and thin structures are subject to more deformation during warping in the image reconstruction process. We believe that addressing these inherent limitations of self-supervised learning is an important step towards increasing its applicability to different tasks and scenarios, and introducing class-specific depth metrics is a useful tool to observe these limitations in the first place. Finally, our qualitative evaluation (cf. Figure 5) clearly highlights that the quantitative performance gains also translate into compelling qualitative improvements.
>
>
> [Consideration should be given to the fact that in a deployment setting, new objects not previously seen in the semantic categories (UFO) may appear.]
>
> The reviewer raises a great point that actually motivated our contribution, based on learning semantic guidance instead of multi-task learning or more explicit semantic priors.
> As stated previously, our depth network is not looking at the final semantic predictions, but rather at intermediate (pretrained) semantic features, which encode appearance information without quantizing it into a set of predefined categories. Generalization out of domain is an open problem in Deep Learning [4], requiring further research beyond the scope of this paper. Nonetheless, our assumption is that something like a UFO does not have to be part of the pre-trained semantic classes, as long as it shares some appearance features with other known categories.
> We find that our depth estimation indeed works well on objects that are outside the semantic segmentation's network ontology, as shown in the added Figure 6 of Appendix B, with objects such as trash cans being correctly reconstructed alongside other erroneous semantic predictions.
>
>
> [1] Vitor Guizilini, Sudeep Pillai, Rares Ambrus, and Adrien Gaidon. Packnet-sfm: 3d packing for self-supervised monocular depth estimation.arXiv preprint arXiv:1905.02693, 2019.
> [2] Huan Fu, Mingming Gong, Chaohui Wang, Kayhan Batmanghelich, and Dacheng Tao. Deep ordinal regression network for monocular depth estimation. In Proceedings of the IEEE Conference on Computer Vision and Pattern Recognition (CVPR), pp. 2002–2011, 2018.
> [3] Clément Godard, Oisin Mac Aodha, Michael Firman, and Gabriel J. Brostow. Digging into self-supervised monocular depth prediction. In Proceedings of the International Conference on Computer Vision (ICCV), 2019.
> [4] Matteo Biasetton, Umberto Michieli, Gianluca Agresti and Pietro Zanuttigh. Unsupervised Domain Adaptation for Semantic Segmentation of Urban Scenes. IEEE Conference on Computer Vision and Pattern Recognition (CVPR) Workshops. 2019

---

> ### Author Response · Authors · 2019-11-12
> **Response to Reviewer #2 (Part 1/2)**
>
> [Table 1 contains results whose significance is hard to judge.]
>
> We agree with the reviewer that the standard depth metrics (abs_rel, RMSE, a1, etc) used by the community [1,2,3] can be opaque, making it difficult to correlate a particular metric with performance on a downstream task. To alleviate this, we propose in our work to also consider a class-specific evaluation, thus allowing further introspection into the performance of our model. This enables us to see improvements in particular classes, that might be more relevant for particular downstream tasks (i.e. road for ground plane extraction, or cars/pedestrians for object detection).
> As for the metrics commonly used to evaluate depth performance, they are helpful in combination because they evaluate depth performance in different ways: Abs.Rel (absolute relative error) and a1 (1.25 distance threshold) roughly measure the overall accuracy of depth estimates, while Sq.Rel. (square relative error) and RMSE (root mean squared error) focus on the variance of depth estimates, being particularly sensitive to outliers. A typical year-over-year relative improvement on these metrics is around 5%, while our method improves by 11% over the previously published state of the art [3], and by 6% over the unpublished strong baseline [1]. Our significant improvements on Sq.Rel. and RMSE indicate that our proposed semantically-guided framework is particularly robust to appearance-related noise. It  generalizes better to different object instances thanks to the introduction of semantic guiding information, where this domain gap is less apparent (i.e. all cars should be treated the same way, regardless of color). This improvement in Sq.Rel. and RMSE can also be explained by the generation of sharper boundaries, as discussed in the paper (and shown in Figure 5), so there are fewer outliers that fall between two objects (a.k.a. the "bleeding" effect).
>
>
> [Table 1 carries no uncertainty in results.]
>
> We agree that, although our trained model is deterministic, there are sources of uncertainty in the proposed approach, as in all learning-based methods. On Table 1, we did not report uncertainty estimates because that is not a common practice in depth evaluation. That being said, we agree with the reviewer. Therefore, we added in Appendix A a study of the main source of uncertainty tied to our major contribution: how different initializations of the semantic network affect the depth fine-tuning process, including semantic training with fewer labels, which results in worse semantic predictions. These experiments show that our proposed framework is robust to the degradation in semantic predictions, which is in line with expectations since we are not using the final semantic predictions themselves, but only guiding our depth network with features from the decoder of the semantic network. This is further illustrated in Appendix B, with examples of situations where the semantic network produces erroneous predictions but the semantically-guided depth network is still able to properly reconstruct objects. This is an indication that the uncertainty encoded in the semantic features is considered within the depth network itself, which learns when semantic information should be used and when it should be discarded to optimize the self-supervised photometric loss.
> In addition, we plan to retrain our whole approach using multiple random seeds to evaluate the end-to-end variance of our method for the final version of the paper, as this takes a significant amount of computational resources and time (hence why no other related work we found does this in practice).

---

### Author Response · Authors · 2019-11-12
**Overview of Responses to Reviewers and Summary of Updated Draft**

We thank all the reviewers for their time and positive feedback regarding our paper's overall clarity, relevance, novelty, and good results, leveraging an implicit connection between semantics and geometry for self-supervised representation learning.

Most of the improvements requested by the reviewers are related to clarifying and improving the robustness of our experimental evaluation. In addition to individual replies to each reviewer (cf. below), we have followed general reviewer recommendations and updated the paper in the following ways:

- Section 5.4.2 (Class-Specific Depth Performance) has been rewritten for clarity. It now properly explains the difference between pixel-average and our proposed class-average depth metrics. We also updated Table 2 and Figure 4, showing clear performance benefits over all semantic classes thanks to both of our contributions (semantic guidance and two stage training).

- We added Appendix A with an analysis on how semantic network pre-training affects our results, as well as the impact of fine-tuning the entire network (semantic + depth). This includes starting from an untrained semantic network and pre-training with a subset of semantic labels, so we can determine how widely applicable our method is, i.e. its robustness to the degradation in semantic guidance (cf. Table 3 in Appendix A and reply to individual reviewers). We find that our method degrades gracefully with worse semantic guidance.

- We added Appendix B with a fine-grained analysis on how erroneous semantic predictions affect our semantically-guided depth network. This further exemplifies how our proposed framework is able to robustly reason over the uncertainty encoded in the semantic features to improve the task of depth estimation.

- We added Appendix C evaluating the generalization capabilities of our semantically-guided framework to a new dataset (NuScenes) that was not available during the pre-training and fine-tuning stages. We show our model generalizes better than our baseline and the state of the art.

Please refer to the replies to individual reviews for more details.

---

### Author Response · Authors · 2019-11-15
**Discussion ends today: is there any remaining unanswered question?**

Dear reviewers, thank you again for your precious time in providing an initial assessment of our contributions. We believe we have satisfactorily addressed most of the points you raised, both in the comments below and in the updated version of our submission. Thanks to your feedback, the submission is now much stronger (quantitatively and qualitatively), with all the requested additional experiments, analyses, details, and clarifications (cf. below). We would like to know if you see some areas that still need improvement, or if you have further questions. We are at your disposal and happy to address any remaining point. Thanks in advance for your additional time and final review.

---

### Decision · Program_Chairs · 2019-12-19

**Decision:**

Accept (Poster)

**Comment:**

The paper proposes a using pixel-adaptive convolutions to leverage semantic labels in self-supervised monocular depth estimation. Although there were initial concerns of the reviewers regarding the technical details and limited experiments, the authors responded reasonably to the issues raised by the reviewers. Reviewer2, who gave a weak reject rating, did not provide any answer to the authors comments. We do not see any major flaws to reject this paper.